# Current and Future Habitat Suitability Models for Four Ticks of Medical Concern in Illinois, USA

**DOI:** 10.3390/insects14030213

**Published:** 2023-02-21

**Authors:** Heather L. Kopsco, Peg Gronemeyer, Nohra Mateus-Pinilla, Rebecca L. Smith

**Affiliations:** 1Department of Pathobiology, College of Veterinary Medicine, University of Illinois Urbana-Champaign, Urbana, IL 61802, USA; 2Illinois Natural History Survey, Prairie Research Institute, University of Illinois Urbana-Champaign, Urbana, IL 61802, USA; 3Institute for Genomic Biology, University of Illinois Urbana-Champaign, Urbana, IL 61802, USA

**Keywords:** ticks, species distribution models, habitat suitability models, Illinois, climate

## Abstract

**Simple Summary:**

The variable landscape of Illinois creates a patchwork of tickborne disease risk to humans and domestic animals that can be predicted in part based on climate and landscape features. We fit individual and mean-weighted ensemble species distribution models for *Ixodes scapularis*, *Amblyomma americanum*, *Dermacentor variabilis,* and a newly invading tick species, *Amblyomma maculatum* using a variety of landscape and mean climate variables and identified numerous environmental niche factors that are associated with the presence of these vectors in current and future climate scenarios within the state. As the environment changes over the coming decades, the distribution of these tick species will change as they adapt to the increasing temperatures and precipitation alterations. Knowing where ticks may concentrate will be important to anticipating, preventing, and treating tickborne disease.

**Abstract:**

The greater U.S. Midwest is on the leading edge of tick and tick-borne disease (TBD) expansion, with tick and TBD encroachment into Illinois occurring from both the northern and the southern regions. To assess the historical and future habitat suitability of four ticks of medical concern within the state, we fit individual and mean-weighted ensemble species distribution models for *Ixodes scapularis*, *Amblyomma americanum*, *Dermacentor variabilis,* and a newly invading species, *Amblyomma maculatum* using a variety of landscape and mean climate variables for the periods of 1970–2000, 2041–2060, and 2061–2080. Ensemble model projections for the historical climate were consistent with known distributions of each species but predicted the habitat suitability of *A. maculatum* to be much greater throughout Illinois than what known distributions demonstrate. The presence of forests and wetlands were the most important landcover classes predicting the occurrence of all tick species. As the climate warmed, the expected distribution of all species became strongly responsive to precipitation and temperature variables, particularly precipitation of the warmest quarter and mean diurnal range, as well as proximity to forest cover and water sources. The suitable habitat for *I. scapularis*, *A. americanum*, and *A. maculatum* was predicted to significantly narrow in the 2050 climate scenario and then increase more broadly statewide in the 2070 scenario but at reduced likelihoods. Predicting where ticks may invade and concentrate as the climate changes will be important to anticipate, prevent, and treat TBD in Illinois.

## 1. Introduction

Ticks and their associated pathogens present a growing public and veterinary health threat in the United States. Human-induced climate and landscape alterations are driving the increased prevalence of emerging tick-borne diseases (TBDs) [1] including bacterial, rickettsial, protozoal, and viral organisms [2,3]. These pathogen emergences are increasingly relevant as the ranges [4,5] and activity periods [4] of native and invasive tick species [3,6] shift, putting these vectors into greater contact with humans, companion animals, and livestock. Economically, changes in tick and TBD ecology are triggering millions of USD in healthcare and livestock impacts [7]. 

Ticks are highly sensitive to and constrained by weather and climate variables [8,9], as well as landscape features such as vegetation and land-use patterns that impact habitat fragmentation [1,10,11]. In general, the questing and phenological activity, development, and survival of common tick species of medical concern are directly correlated with higher levels of humidity and warmer temperatures [8,12,13,14]. However, these impacts are species-specific. Ticks such as *Ixodes scapularis* are highly susceptible to desiccation, whereas *Amblyomma americanum, Amblyomma maculatum*, and *Dermacentor variabilis* are more tolerant of drier conditions [9,15]. Greater tick density is often associated with habitats that include uninterrupted forest cover [16,17], or even specific invasive types of landscape cover [18], but edge effects and open landscapes can also foster high tick abundance depending on species [15,19]. These landcover and climate relationships are critical to the landscape epidemiology of TBD because they generate the microclimatic conditions that facilitate interactions among ticks and their hosts [1,20]. 

The greater U.S. Midwest is on the leading edge of tick and TBD expansion. Within the past decade, studies have documented the continued range movement of four ticks of medical and veterinary concern in this region including the blacklegged tick (*I. scapularis*) [21,22], lone star tick (*A. americanum*) [23,24], American dog tick (*D. variabilis*) [25,26], and Gulf Coast tick (*A. maculatum*) [19,22,27,28]. These range expansions have corresponded with an increase in reported TBD cases associated with these species including Lyme disease [29], ehrlichiosis [30], tidewater fever [27], and the newly documented Heartland virus [31]. 

Illinois is experiencing tick and TBD expansion in both the northern and the southern regions [23,32,33,34]. Concurrently, there has been a 10-fold increase in commonly reported TBD cases among humans between 1999 and 2017 [35,36,37,38], including Lyme disease, Rocky Mountain spotted fever, ehrlichiosis, and anaplasmosis. Three distinct climate regions exist longitudinally across the state [39], with clear impacts on tick species abundance [9]. As climate alterations impact the various bioclimatic factors across these areas, it is important to predict how the tick distribution and TBD risk will potentially change across the state.

Although there is debate about the specific impacts of extreme climate conditions on ticks and TBDs in the future [40], climate projection models can predict and assess various current and future habitat and distribution scenarios. Species distribution models (SDM) represent a suite of statistical and machine-learning methods for predicting suitable species habitat ranges and niches based on known occurrence records and various environmental variables. These strategies range from deterministic (e.g., logistic regression) to stochastic (e.g., Bayesian regression trees) approaches, and utilize various levels of model validation techniques. Given the differences in model performance, using the SDM model ensembles may provide a more complete picture of the possibilities for tick species range variation, and opportunities for public health and veterinary partners to enact control and prevention measures where most needed [41,42]. 

The objective of this study was to fit and evaluate current and future species distribution models for each of the four tick species of major medical and veterinary concern within Illinois, including *I. scapularis*, *D. variabilis*, *A. americanum*, and *A. maculatum,* and to evaluate habitat and climate variables associated with their predicted occurrence. We expected that as the climate continues to warm, habitats in southern and central Illinois will become less hospitable for a desiccant-sensitive species such as *I. scapularis* but more habitable for the other three more desiccant-tolerant species. This hypothesis would reflect a greater predicted species range throughout the state for the *Dermacentor* and *Amblyomma* species but would result in a growing absence of suitable *I. scapularis* habitat, except in the northernmost part of the state.

## 2. Materials and Methods

### 2.1. Tick Occurrence Data

We sourced presence-only tick occurrence records from several online, publicly accessible databases and through active tick collections throughout Illinois. The databases included Walter Reed Biosystematics Unit (WBRU)’s VectorMap [43] (http://vectormap.si.edu/; with data from WRBU and the Illinois Natural History Survey Insect Collection; accessed 8 June 2022), Global Biodiversity Information Facility [44] (GBIF; https://www.gbif.org/; with data from iNaturalist, Canadian Museum of Nature, Chicago Academy of Sciences, Harold W. Manter Laboratory of Parasitology Collection; accessed 9 June 2022), and Biodiversity Information Serving Our Nation [45] (BISON; https://bison.usgs.gov/; with data from BISON, iNaturalist, and the Illinois Natural History Survey Insect Collection; accessed 9 June 2022). To be included in a model, all tick occurrence data had to meet the following quality control criteria: be an observation from no earlier than 1950, include two decimal places or more for at least one coordinate, and have a coordinate inaccuracy of ≤20,000 m. Duplicate coordinates occurred often due to data being deposited across multiple databases, so entries were compared and duplicate coordinates were removed. Geolocations were cross-checked to ensure that records were accurate to the field location. The remaining coordinates were then thinned to a 1 km distance using the spThin package [46] to reduce the effect of sampling bias on model predictions.

### 2.2. Environmental Covariates

WorldClim 2 bioclimatic variables (1–19) [47] (Table 1) were sourced from the *geodata* package [48] and downloaded at a resolution of 0.5 arcminutes (~1 km^2^). Current climate models were fit using the historical data representing the average measurements from 1970 to 2000. Future climate models were fit with mean projections of these data at a ~1 km^2^ resolution using a Coupled Model Intercomparison Project Phase 6 (CMIP6) Earth Systems Model (EC-Earth3-Veg) [49] under Shared Socioeconomic Pathways (SSP) 585 for 2050 (average from 2041 to 2060) and 2070 (average from 2061 to 2080). Overall, ESMs such as EC-Earth3-Veg tend to describe the most relevant climate feedback mechanisms and provide more thorough uncertainty measurements than global circulation models (GCMs) [50]. Whereas using an ensemble of future climate models is generally employed when fitting SDMs for larger regions to minimize individual model bias, we chose a single ESM that is shown to perform best for the small region we are modeling [51]. Using an ensemble that incorporates numerous GCMs/ESMs over a small region like a single state can skew predictions [51]. SSP 585 is a future climate scenario that describes the expected baseline high greenhouse gas impact resulting from a lack of carbon emission mitigation policies [52], i.e., a “worst case” scenario.

Due to the importance of white-tailed deer (*Odocoileus virginianus*) as reproductive hosts for each of these four species, we included a raster of suitable deer habitats within Illinois [53]. Landcover class (Figure 1) and percent impervious surface from the National Land Cover Database (NLCD) [54] were also included. The NLCD is a collection of land cover imagery at 30m resolution that combines information from all years of land cover change (2001-2019) across 16 classes of cover that include impervious land, cropland, wetland, and various vegetation types, which were aggregated into seven more general land cover categories (water, developed, barren, forest, grass/shrub, cropland, and wetland). An average of these land cover classes was taken every 2–3 years instead of data from a single year to adjust for the change that occurred from 2001 to 2019. The elevation was sourced from the *raster* package [55] derived from Shuttle Radar Topography Mission (SRTM) National Elevation Dataset digital elevation models (at a resolution of 1 and 1/3 arcseconds) [56].

All covariates were cropped to the extent of Illinois’ state borders (xmin: −91.5°, xmax: −87.5°, ymin: 36.9°, ymax: 42.5°) and resampled to a resolution of 1 km (0.5 arcminutes) to match the bioclimatic datasets for the specific climate projection period. Extracted covariate values were assessed for collinearity for each species and period by assessing the variance inflation factor. Any variable with a v-step score of 10 or higher was excluded from that species and climate period model due to collinearity.

### 2.3. Model Fitting and Evaluation

Models were fitted using the *sdm* package [57] in R version 4.1.3 [58]. Regression and machine learning models for each species for the current climate were first fitted using the following individual methods: generalized linear models (GLM), generalized additive models (GAM), Bayesian regression trees (BRT), classification and regression trees (CART), MaxEnt, random forest (RF), multivariate adaptive regression splines (MARS), and support vector machines (SVM). The number of randomly selected pseudo-absence points was set at approximately the same number of presence points for each species due to the mixed use of regression and machine learning techniques within the modeling algorithm [59] and was also thinned to 1 km^2^ to match the presence points. Cross-validation and bootstrap data partitioning methods (with a 30% test percentage) were used for each model type, with five replicates for each method totaling five replicates per algorithm (30 total replicates per species). Single model algorithms that were not 100% successful during replicate runs were excluded from ensemble models. Models were evaluated using several performance scores including threshold-dependent and threshold-independent methods: area under the curve (AUC), true skill statistic (TSS), model deviance (DEV), and correlation (COR). Single models demonstrating AUC > 0.75 and TSS > 0.50 were retained for mean-weighted ensemble models (i.e., a two-step process that incorporates both within-model averaging and between-model averaging). Cohen’s kappa was not used for single model evaluation due to its overreliance on prevalence but was consulted to determine consistency in predictions across models [60]. AUC was not used alone to assess the prediction accuracy because of its poor ability to reliably assess the presence-background nature of the tick occurrence data [60,61].

## 3. Results

### 3.1. Ixodes scapularis Models

After duplicate records were removed and presence points thinned there remained 62 known *I. scapularis* occurrence points across Illinois, and 70 pseudoabsence points were randomly generated (Table 2). After assessing for multicollinearity amongst environmental variables for all climate periods, fifteen predictor variables out of the 29 total environmental covariates were removed from the dataset due to v-step scores greater than 10 (bio1, bio2, bio3, bio4, bio6, bio10, bio11, bio12, bio14, bio15, bio16, bio17, bio19, percent impervious surface, percent white-tailed deer habitat). Retained in the historical, 2050, and 2070 climate correlate dataset for *I. scapularis* were bio5, bio7, bio8, bio9, bio13, bio18, elevation, percent water body coverage, percent barren land, percent forest, percent grassland, percent cropland, and percent wetland.

As the algorithm evaluation revealed RF to be the best-fit model for predicting the historical climate distribution of *I. scapularis* (Table 2). The landscape variables that most strongly predicted the occurrence of *I. scapularis* habitat across this model in the historical climate were percent forest (16.5% relative contribution), percent wetland (10.8%), and percent grassland (11.1%). Climate variables all contributed less than 5% each. *I. scapularis* was predicted to occur above 80% likelihood in forest landscapes with less than 50% coverage and was expected to increasingly occur with greater percentages of wetlands and grasslands. Predicted *I. scapularis* occurrence was less likely with increasing maximum temperatures in the warmest month (bio5), increasing mean temperature of the driest quarter (bio9), increasing annual temperature range (bio7), and increasing precipitation in the wettest month (bio13). *I. scapularis* was more likely to occur with increasing precipitation in the warmest quarter (bio18), and with increasing temperatures (up to 21 °C) in the wettest quarter.

The best fit single algorithm models for future predictions were RF (2050; AUC = 0.82, COR = 0.55, TSS = 0.64, DEV = 1.08) and SVM (2070; AUC=0.90, COR = 0.72, TSS = 0.76, DEV = 0.81). Percent forest coverage became more important in predicting the likelihood of *I. scapularis* in the 2050 and 2070 climate scenarios, increasing to 27.2% (2050) and then 28% (2070) relative contribution (Table 3). By 2070, the wetland percentage rose to 13.6% relative contribution and was one of the most important variables along with the precipitation of the warmest quarter (bio18) (Table 3). Overall, *I. scapularis* displayed the same response to the landcover and climate variables in the future scenarios as it did in the historical climate. As the climate changed, the difference that occurred was that the presence of *I. scapularis* was increasingly more likely in habitats with greater forest cover (between 50% and 75% coverage), and the overall greatest predicted probability of *I. scapularis* for any variables dropped to approximately 85% likelihood in 2050, and then to 75% in 2070.

Best fit mean-weighted ensemble models for both historical and future climate scenarios included the following algorithms: GLM, BRT, CART, MaxEnt, RF, and SVM. Within the historical climate, the best fit ensemble models predicted that *I. scapularis* would most likely be found within the Chicago metropolitan statistical area (CMSA) along the northeastern border of Lake Michigan, along riparian zones in western and central Illinois, and within the forested regions of east-central and southern Illinois (Figure 2a). The tick species was also expected to be found scattered throughout forested pockets within the central portion of the state. As the climate warmed in the 2050 (Figure 2b) and 2070 (Figure 2c) projection scenarios, the likelihood of *I. scapularis* presence throughout the central and southern tiers began to recede and concentrate along rivers and waterbodies (2050), and then shifted to a greater expectation of occurrence within the heavily forested region of Southern Illinois (2070) (Figure 3).

### 3.2. Amblyomma americanum Models

After removing duplicate observations and occurrence points were thinned to 1 km, 99 records of *Amblyomma americanum* were retained for modeling and 100 randomly selected pseudoabsence points were generated (Table 2). Fifteen variables were removed due to multicollinearity (bio1, bio3, bio4, bio5, bio6, bio7, bio10, bio11, bio12, bio14, bio16, bio17, bio19, percent developed land, and percent impervious surface,). Retained for the modeling of all climate periods were bio2, bio8, bio9, bio13, bio15, bio18, and land cover categories elevation, white-tailed deer habitat, percent water body coverage, percent barren land, percent forest coverage, percent grassland, percent cropland, and percent wetland.

Random forest was the best fit single model algorithm for the predicted *A. americanum* habitat distribution of the six total included model algorithms across all climate periods (Table 2; 2050: AUC = 0.89, COR = 0.70, TSS = 0.74, DEV = 0.84; 2070: AUC = 0.92, COR = 0.74, TSS = 0.76, DEV = 0.76). The most important variables that predicted the occurrence of *A. americanum* habitat across this model for the historical climate were percent wetland (17.9% variable contribution), percent forest coverage (11.8%), and the presence of white-tailed deer habitat (3.9%). Climate variables bio2 (1.8%) and bio15 (2.4%) were the most important contributing climate variables to the historical climate prediction of *A. americanum* distribution (Table 2). The presence of *A. americanum* was between 80 and 90% likely to occur in forest habitats with less than 50% coverage in the historical climate period. Under these conditions, *A. americanum* is also positively associated with barren land, white-tailed deer habitat, and wetland landcover, with occurrence probabilities as high as 90% with increasing percent class coverage. The likelihood of occurrence declined slightly with increasing grassland and waterbody coverage. The probability of this species’ occurrence briefly increased and then sharply declined with the increasing percentage of cropland coverage, precipitation seasonality (bio15), precipitation in the warmest quarter (bio18), mean diurnal range (bio2), and mean temperature of the wettest quarter (bio8). The occurrence of *A. americanum* was negatively associated with increasing precipitation of the wettest month (bio13) and was positively associated with increasing temperature of the driest quarter (bio9). In future climate scenarios, *A. americanum* responded to these land cover and climate variables in the same way except that this tick was more likely to occur in higher percents of cropland and forest and was more significantly predicted by mean diurnal range (bio2) than in the historical climate period. In the 2070 mean climate, the overall probability of *A. americanum* occurrence across all variables was lower.

The best fit mean-weighted ensemble models for both current and future climate scenarios included the following algorithms: GLM, BRT, CART, MaxEnt, RF, MARS, and SVM. Historical *A. americanum* distribution was predicted to be greatest in the southern-most portion of the state where there is more contiguous forest and suitable white-tailed deer habitat, along riparian zones of the Illinois and Rock River systems, and within the Chicago metropolitan statistical area (Figure 4a). As the climate scenarios progress to the 2050 mean climate, *A. americanum* appears to occur with a greater likelihood in Shawnee National Forest but is less likely to occur in the white-tailed deer habitat between forest patches in and around Lake Shelbyville and the Kaskaskia River, the Illinois River, the forest to the west, the Rock River, and within the CMSA (Figure 4b and Figure 5). By 2070, *A. americanum* is expected to occur more broadly within croplands across the state, but at lower probabilities. It is also less likely to occur in the CMSA (Figure 4c and Figure 5).

### 3.3. Dermacentor variabilis Models

After removing duplicate records and thinning observations, 290 records of *D. variabilis* were retained for modeling, and 300 randomly generated pseudoabsence points were generated. Best fit models for all climate conditions for *D. variabilis* included bio2, bio7, bio8, bio9, bio10, bio13, bio18, elevation, percent water body coverage, percent barren land, percent developed land, percent forest, percent grassland, percent cropland, and percent wetland. Fourteen covariates (bio1, bio3, bio4, bio5, bio6, bio11, bio12, bio14, bio15, bio16, bio17, bio19, percent impervious surface, and suitable white-tailed deer habitat) were removed from consideration in the historical climate model due to collinearity issues.

Random forest was the best fit single-model algorithm for predicting the presence of *D. variabilis* within the historical and both future climate periods (Table 2; 2050: AUC = 0.89, COR = 0.68, TSS = 0.67, DEV = 0.85; 2070: AUC = 0.88, COR = 0.68, TSS = 0.67, DEC = 0.86). The percent forest coverage was the most important variable in predicting *D. variabilis* occurrence (30.4% relative importance) in the historical period, followed by percent wetland (12.7%) (Table 3). The climate variable that contributed most to the model was mean diurnal range (bio2; 8.6%) (Table 3). A greater likelihood of *D. variabilis* presence was predicted with increasing percentages of barren land and wetland. *D. variabilis* was 90% likely to occur in habitats with up to 25% forest coverage, but decline to near 50% occurrence likelihood as forest percentage increased. Similarly, *D. variabilis* was less likely to occur with increasing cropland, developed land, grassland, and water body percentage. The predicted occurrence of *D. variabilis* increased to and then remained at 75% likelihood in response to the mean temperature of the driest quarter (bio9), and its presence declined with the increasing mean temperatures of the warmest quarter (bio10). In response to the precipitation of the wettest month (bio13), the precipitation of the warmest quarter (bio18), the mean diurnal range (bio2), the temperature annual range (bio7), and the mean temperature of the wettest quarter, *D. variabilis*’ predicted occurrence initially increased with the increasing temperature or precipitation, but then declined to near 50% probability.

For the 2050 climate projection, percent forest (30.4% relative contribution) and wetland (12.3%) were again the most important variables in predicting the occurrence of *D. variabilis,* followed by elevation (3.8%) and mean diurnal range (bio2; 3.4%) (Table 3). As the percentage of forest increased above 25%, the expected occurrence of *D. variabilis* declined from nearly 90% to approximately 60% likelihood. Under the conditions expected in this period, this species was more likely to be found in areas with water bodies (between 80 and 85% likelihood). The expected occurrence of *D. variabilis* followed the same increase followed by decrease trends with climate variables as it did in the historical period, but the overall percent likelihood of occurrence reduced from the 90-60% range to the 80-50% range. Similar response patterns are seen in the 2070 projected climate compared to the 2050 climate, but with overall less expected occurrence of *D. variabilis*, and a more dramatic response to the changing variables. Increasing the percentage of forest reduces the overall likelihood of *D. variabilis* to 40% at 100% forest coverage. Percent barren land continued to predict *D. variabilis* presence consistently at 75% probability. The likelihood of *D. variabilis* was positively associated with increasing precipitation of the wettest month (bio13) and precipitation of the warmest quarter (bio18), but only until 300 mm, after which, it declined. The likelihood of *D. variabilis* occurrence declined to below 50% when the mean diurnal range increased above 11 °C and declined to a 60% probability of occurrence when the temperature annual range (bio7) rose above 40 °C.

Best fit mean-weighted ensemble models for historical and future climate scenarios included the following algorithms: GLM, BRT, CART, MaxEnt, RF, MARS, and SVM. The occurrence of *D. variabilis* under historical climate conditions was predicted to be distributed throughout the state, with concentrations of higher probability located within southern Illinois, the CMSA, and along riparian zones (Figure 6a). All climate scenarios predict that the probability of *D. variabilis* occurrence is generally resilient in most habitats except cropland but is increasingly dependent on a lower temperature and higher precipitation as the climate shifts into the more extreme 2070 projections (Figure 6b,c). By 2050, the tick species is less likely to occur along the northern border of the state, as well as along riparian zones of the Illinois and Kaskaskia Rivers (Figure 7). By 2070, the tick appears to increase in likelihood within those areas but becomes less likely to occur in the Shawnee Forest region of southern Illinois and the south-central forested region. (Figure 7).

### 3.4. Amblyomma maculatum Models

Fifteen records of *A. maculatum* were retained for modeling after removing duplicates and thinning and were combined with 20 randomly selected pseudoabsence points. A total of twenty environmental correlates were removed due to multicollinearity (bio1, bio2, bio3, bio4, bio5, bio6, bio7, bio8, bio9, bio10, bio11, bio12, bio13, bio14, bio16, bio17, bio19, percent white-tailed deer habitat, elevation, and percent impervious surface). The remaining predictors in each climate model for *A. maculatum* were bio15, bio18, percent water bodies, percent barren land, percent developed, percent forest, percent grassland, percent cropland, and percent wetland.

Support vector machines were the best fit model to predict the historical and 2070 distribution of *A. maculatum* (Table 2; 2070: AUC = 0.77; COR = 0.47; TSS = 0.67; DEV = 1.24), whereas RF was the best algorithm for the 2050 climate period (AUC = 0.75, COR = 0.48, TSS = 0.65, DEV = 1.23). Percent forest (29.1% relative importance), percent developed land (10.3%), percent wetland (10.1%), and percent cropland (10.0%) were the most important landscape variables in predicting the probable locations of *A. maculatum* during the historical and 2050 climate periods (Table 3). In 2070, percent barren land and grassland also become more important predictors of *A. maculatum* occurrence (Table 3). Precipitation seasonality (bio15) and precipitation of the warmest quarter (bio18) were the only significant climate variables included in the models, and these both increased in relative contribution to the model in the 2050 and 2070 scenarios (Table 3). In the historical climate, the occurrence of this tick species was expected in open landscapes, and was positively correlated with increasing percentages of grassland, cropland, and developed land, and negatively associated with forests. Proximity to waterbodies and wetlands also increased the probability of *A. maculatum* occurrence during this period. Variation in precipitation as well as total precipitation in the warmest quarter initially was associated with an increase in the likelihood of *A. maculatum* occurrence (to only a 50-55% likelihood), but then likelihood decreased as the coefficient rose above 25 and as precipitation in the warmest quarter reached 330mm.

As the climate warmed in 2050 and 2070, the occurrence probability of *A. maculatum* shifted to specific habitats. Percent forest, developed land, and precipitation of the warmest quarter (bio18) became the most important variables for the model in 2050 (50.0%, 6.2%, and 7.8% relative importance, respectively). The probability of *A. maculatum* occurrence increased in areas with a greater percentage of developed landcover (i.e., CMSA) and decreased in response to an increasing percentage of forest. Precipitation of the warmest quarter (bio18) was associated with a sharp increase in the likelihood of *A. maculatum* occurrence during this time period, and increasing seasonality of precipitation (bio15) was associated with an increase in the probability of occurrence to approximately 30mm, and then a sharp decline in the probability of occurrence. Under the 2070 climate scenario, percent forest contributed slightly less to the *A. maculatum* distribution model (45.3%) as other variables increased in relative contribution. Percent wetland, percent barren land, percent cropland, percent grassland, and percent developed land contributed roughly the same amount to the model (13.1%, 9.8%, 8.6%, 7.7%, and 7.6% relative importance, respectively; Table 3). Precipitation of the warmest quarter (bio18) was also a driving climate variable in the prediction of *A. maculatum* distribution (10.4% relative variable importance). Within these future conditions, *A. maculatum* was predicted (between 40 and 70% likelihood) to be associated with open landscapes like increasing barren land, cropland, developed land, and grasslands. The occurrence of this tick species was also expected at a 40-50% probability in areas with 10-20% water bodies and wetlands. As the variation in precipitation across seasons (bio15) increased above 15 mm, the likelihood of *A. maculatum* occurrence decreased from a 45% likelihood to a below 35% probability. Increasing precipitation of the warmest quarter (bio18) was associated with a strong increase in *A. maculatum* occurrence probability from 30% to 60% as precipitation reached 340 mm.

These model predictions suggest that *A. maculatum* has wide distribution potential throughout Illinois in the historical climate scenario. It is most likely to be able to survive in open barren landscapes that are close to water sources and wetlands, placing the most probable distribution predictions along the Illinois and Rock rivers, and around the CMSA on the banks of Lake Michigan. Scattered pockets of higher probability throughout the state correspond with areas that are devoid of dense forest (greater than 50% coverage) or areas that are more than 50% cropland (Figure 8a). The 2050 (Figure 8b) and 2070 (Figure 8c) climate prediction ensemble models projected that *A. maculatum* distribution would reduce overall, with covariates only predicting the likelihood of tick occurrence as high as 70%. During these scenarios, *A. maculatum* was generally more prevalent in areas with less than 50% forest cover. In 2050, the distribution of *A. maculatum* was predicted to be more highly concentrated in open habitats near rivers and the floodplains of water bodies, as well as an increased probability of occurrence in the CMSA (Figure 8b). The predictions for *A. maculatum* distribution in the 2070 scenario appear to change drastically, with a higher likelihood of occurrence throughout the central portion of the state—particularly in areas with a high percentage of cropland—and less strongly associated with wetlands (Figure 8c). Change over time shows a south-to-north shift in suitable habitats across the state (Figure 9).

## 4. Discussion

This investigation applied numerous species distribution modeling techniques to examine the historically predicted distribution of four ticks of medical concern in Illinois, and the estimated future habitat suitability based on two climate scenarios. With the exception of *A. maculatum*, our results support known [33,62,63,64] and predicted [19,65,66,67] habitat ranges for these species within the state and attempt to identify environmental factors that will contribute to continued or altered suitability distributions in potential future climate conditions.

The best fit individual models to describe these historical and future habitat suitability scenarios were random forest and support vector machines. Random forest is a specific type of classification/regression tree (CART) ensemble and recursive partitioning method that can handle highly dimensional data with accuracy and is resistant to overfitting due to its randomized splitting and sampling procedure of training data [68]. However, Valavi et al. [69] note that RF prediction can be negatively impacted when using presence-background data such as the tick data used in this investigation due to class imbalance and overlap. This occurs when there is a small sample of presence points, and the background points are sampled in a way that does not allow for enough discrimination in the predictors of presence and background location [69]. Support vector machines are another machine learning algorithm that utilizes kernel function for mapping presences amidst complex correlational data, and they are useful because they do not require data to be independent [70,71]. To prevent biasing the outcomes as best as we could, we applied mean weights and approximate equal sampling of the presence and the background data and used down-sampling by way of cross-validation [69]. The best-performing models were consistently the mean-weighted ensembles, which is an outcome supported by previous research [71].

Our models support and expand upon previous work on habitat suitability for ticks in Illinois. Records of county-level establishment, passive surveillance, and ecological niche modeling demonstrate the expansion of *I. scapularis* across the state [34,67,72,73,74]; however, our expectation that as the climate continues to warm, regions in southern and central Illinois will become less hospitable for a desiccant-sensitive species such as *I. scapularis* was not supported. We found that the occurrence of these species is still driven strongly by precipitation and temperature variables as previous work in the state has demonstrated [9] but in ways we did not expect. Our models predicted that *I. scapularis* will initially be confined to more northern regions in the state, and within habitats that provide more protective cover (e.g., upland forest) and moisture availability, e.g., along riparian zones of the Sangamon, Rock, and Illinois Rivers, as well as in forested areas and edge surrounding Lake Shelbyville and Upper Peoria Lake. However, Shawnee National Forest becomes highly likely to be suitable refugia in the 2070 climate scenarios.

We observed the potential continued future suitability of habitat for *I. scapularis* located in high population centers such as Cook, DuPage, McHenry, and Lake Counties outside Chicago. *Borrelia burgdorferi*-infected *I. scapularis* has been collected from high-access areas within these locations going back decades [62,63]. Guerra et al. [65] identified positive associations of *I. scapularis* with various soil types (e.g., fertile alfisols, sand, and loam), deciduous and dry forests, and negative associations with grasslands, acidic soils, conifer, and wet forests. At that time, highly likely (>0.50) habitat suitability for *I. scapularis* was largely limited to areas within Shawnee National Forest, along the Illinois and Mississippi Rivers, and very few areas of higher probability of presence (0.50-0.75) in the counties surrounding Chicago [65]. We predicted greater suitability for *I. scapularis* throughout the central and southern portions of the state than what was previously predicted or currently reported by Illinois Department of Public Health records [75]. It is suspected that *I. scapularis* is not currently occupying a larger distribution within Illinois due to its complex ecology [1,76,77]. Although our results could suggest that the tick simply has not yet invaded these areas, they may also reflect sampling limitations. Lyons et al. [38] found few *I. scapularis* ticks during active surveillance in southern Illinois, but the timing of this surveillance was not optimized to the phenology of *I. scapularis*, and passive surveillance efforts lacked coverage in many areas of interest.

Levi et al. [78] examined activity patterns of *I. scapularis* over 19 years and found that years with warmer temperatures in the summer and fall were associated with a three-week acceleration in the phenology of nymphal and larval ticks as compared to years with lower temperatures. Model predictions suggest up to a two-week average earlier activity period for larvae and nymphs if 2050 warming predictions hold [78], which provide additional opportunity for overlap with humans and domestic animals. Given that the risk of acquiring a tickborne illness such as Lyme disease is heavily dependent on not only the enzootic cycle of disease but also on human behavior, our predictions can help identify areas of Illinois to concentrate additional surveillance efforts to more accurately quantify that acarological risk.

The predicted *A. americanum* habitat for the historical climate closely matches reported occurrences within Illinois [23,34,75]. Currently, this species is most abundant in the southern portion of the state but is becoming increasingly more common in the north [24,79,80]. *A. americanum*’s aggressive host-seeking and non-specific host preferences create an optimal dispersal scenario which allows this tick to travel long distances on meso-mammals and deer, as well as birds [81]. Despite all of the modeled tick species relying on white-tailed deer for at least part of their lifecycle, *A. americanum* was the only species distribution significantly predicted by suitable white-tailed deer habitat. We suspect that this is because white-tailed deer (*Odocoileus virginianus*) are the preferred hosts for all life stages of *A. americanum* [82], and their distribution has historically been intricately tied to the presence of white-tailed deer throughout their eastern U.S. range [79,83].

However, *D. variabilis* and *A. americanum* were also found to be constrained by the 2070 climate scenarios in similar habitats but were more likely to occur throughout more of the forested southern portion of the state, like previous research [84]. *A. maculatum* was the only tick predicted to continue to expand throughout the state as the temperatures rose more extremely into the 2070 climate. Of greatest concern for public health is the increasing likelihood of these additional vector tick species near the higher population centers along the Illinois River and surrounding Chicago. We have found that few medical professionals in northern areas of Illinois were familiar with the risk of ehrlichiosis within the state [85], despite 422 cases between 2011 and 2021 [75]. This is likely due to the current abundance of *A. americanum* being higher in the southern portion of the state and is likely to delay the diagnosis and treatment of pathogens vectored by these less-studied tick species. Both Bayles et al. [86] and Soucy and de Urioste-Stone [87] also found that the adoption of effective tick prevention measures, such as tick checks, was associated with the perceived risk of tick bites. As the actual risk of tick exposure changes due to shifting tick habitats and abundance, public and professional awareness must be addressed through dynamic communication efforts.

We focused on more extreme expectations for future climate scenarios to capture a likely “worst case scenario” for future tickborne disease risk, mainly because the entirety of Illinois is expected to be within a projected “extreme heat belt” with heat index temperatures exceeding 125 degrees Fahrenheit for at least one day by 2053 [88]. Broader studies that have examined potential tick niche expansion and retraction under future scenarios have found similar results for these species regardless of the global circulation model chosen. Ma et al. [79] explored the impact of several shared socioeconomic pathways through 2100 and predicted all of Illinois to be highly suitable for *A. americanum* during all scenarios ranging from least impactful to most impactful. These projections combine climate model data with policy to best capture a likely outcome for climate change. Employing ecological niche models with future greenhouse gas emission scenarios such as Representative Concentration Pathway (RCP) 4.5 (moderate warming) and RCP 8.5 (severe warming) predicted a similar suitability outcome for *A. americanum* in North America [89,90], as did a study by Boorgula et al. [25], which predicted moderate to high suitability for *D. variabilis* throughout the state continued through both RCP 4.5 and RCP 8.5 scenarios. Although Flenniken et al. [19] did not examine future projections of *A. maculatum*, they found that under current climate conditions, the expected ecological niche for this species is much greater than its current distribution, suggesting the potential for expansion north and east. By focusing on Illinois alone, we were able to apply a more fine-scale environmental niche prediction for each of these four tick species within the SSP 585 scenario.

We recognize several limitations in our investigation. It is important to note that species distribution modeling is often subject to confounding due to the phenomenological approach to predicting tick distributions. Spurious correlations can be assumed without an additional mechanistic understanding of the relationships between ticks and these environmental predictors at a smaller scale [14]. We attempted to control for this, in part, by including known white-tailed deer habitats, but this variable was removed due to collinearity issues with other environmental correlations for all tick species except for *A. americanum*. Previous work also demonstrated that for certain species, such as *I. scapularis*, tick presence varied despite host availability, suggesting a more influential role of abiotic variables [65]. Whereas the inclusion of other forest-level habitat variables likely replaced the need for specific deer habitat for three of the four ticks, we consider it a limitation given the need for and importance of considering reproductive host species in habitat models. However, in the case of the Lyme disease bacteria (*Borrelia burgdorferi*), recent evidence may suggest that the overall tickborne disease hazard risk posed by the positive association between deer density and nymphal tick density is canceled out through opposing forces of both amplification and dilution since deer are not a competent reservoir for the bacteria [91].

Our environmental correlates included climate variables that change according to proposed scenarios, but the landcover predictors did not include estimates of variability. As landcover predictors did not change over time, our model results, therefore, assume that the changes in climate do not change the percentage of cropland or other landcover types throughout the state. ESRI landscape change predictions for 2050 in Illinois included an expected gain of over 821,000 acres of cropland throughout the state, a gain of over 503,000 acres of developed or impervious surface, and losses of deciduous forest (743,000 acres), grassland (380,000 acres), and wetland (39,224 acres) [92]. Future modeling work should include these predictions to improve upon static landscape assumptions. Further, the historical climate and landscape variables were slightly mismatched (climate was a mean from 1970 to 2000, whereas the landscape mean ranged from 2001 to 2019). These differences could potentially impact the model’s accuracy. We did not incorporate soil types or profiles [65] into the models which also may have impacted the predictions due to their ability to harbor and control microclimates and habitats that can impact tick survival. However, since only certain vegetation is expected to grow according to various soil profiles [65], we assumed that vegetation was enough of a proxy for these models.

Booth [93] reported that certain combined temperature and precipitation bioclimatic variables can be unreliable in species distribution modeling depending on proximity to the equator due to discontinuities in interpolation and can result in extreme differences over short distances. In the United States, specifically, the mean temperature of the warmest quarter (bio8) and the mean temperature of the driest quarter (bio9) demonstrated anomalies in the south and southeastern regions of the country [93]. These discontinuities were like others that occurred globally near the equator. These anomalies should not have impacted our results because of Illinois’ distance from the equator, but mention is warranted since these variables were important in our models.

Sampling bias consideration is important with occurrence data and may have influence potentially seen in the response curves of *I. scapularis* in the historical climate. Previous research [1] showed an increasing likelihood of the presence of *I. scapularis* in uninterrupted forests, whereas our results demonstrate a large decline in the likelihood of *I. scapularis* occurrence with an increasing percentage of forest cover. This could reflect a lack of data points collected from deeper within forests (i.e., collections were intentionally performed in easily accessible places because this is where the disease transmission risk is), or that this species spends more time in edge environments within Illinois. The sampling method (drag versus CO2 trap versus small animal capture) is also important to consider when assessing bias. Rynkiewicz and Clay [15] reported that *I. scapularis* was mainly found collected from small mammals, whereas *A. americanum* and *D. variabilis* were able to be collected using cloth drag and CO2 protocols. Records of *I. scapularis* in Illinois may therefore be underrepresented, as most sampling in the state has used the cloth drag approach.

The very small dataset for *A. maculatum* may have contributed to the projected future results suggesting a lack of *A. maculatum* in landscapes that it is known to thrive in, such as grasslands, or future projections associating the tick with croplands. Specifically, the sample size may have impacted the accuracy of the random forest/CART predictions per class overlap as previously stated [69]. The reevaluation of this tick’s expected distribution as more data become available is necessary.

## 5. Conclusions

The variable landscape of Illinois creates a patchwork of risk to humans and domestic animals that can be predicted based on climate and landscape features. As the climate changes over the coming decades, the distribution of these tick species will change as it adapts to the increasing temperatures. Knowing where ticks may concentrate will be important to anticipating, preventing, and treating tickborne disease.

## Figures and Tables

**Figure 1 insects-14-00213-f001:**
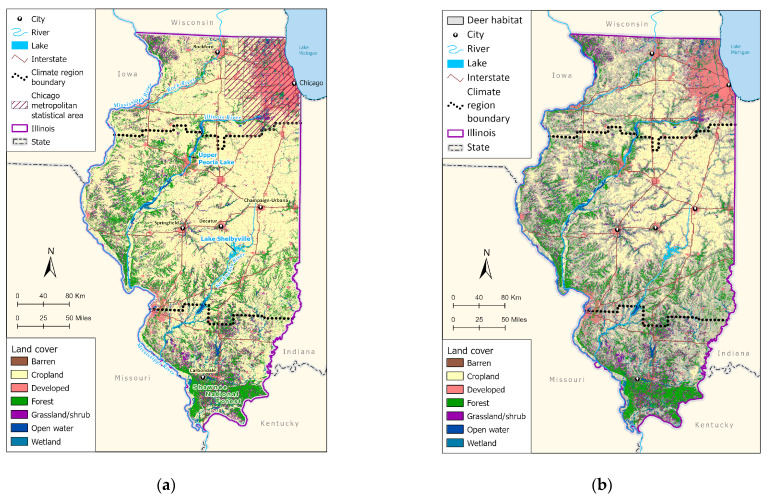
(**a**) National Land Cover Database [54] aggregated landcover classes for Illinois. Climate region boundaries are derived from the National Oceanic and Atmospheric Administration U.S. Climate Divisional Dataset [39]. (**b**) NLCD aggregated land cover with white-tailed deer habitat overlay [53].

**Figure 2 insects-14-00213-f002:**
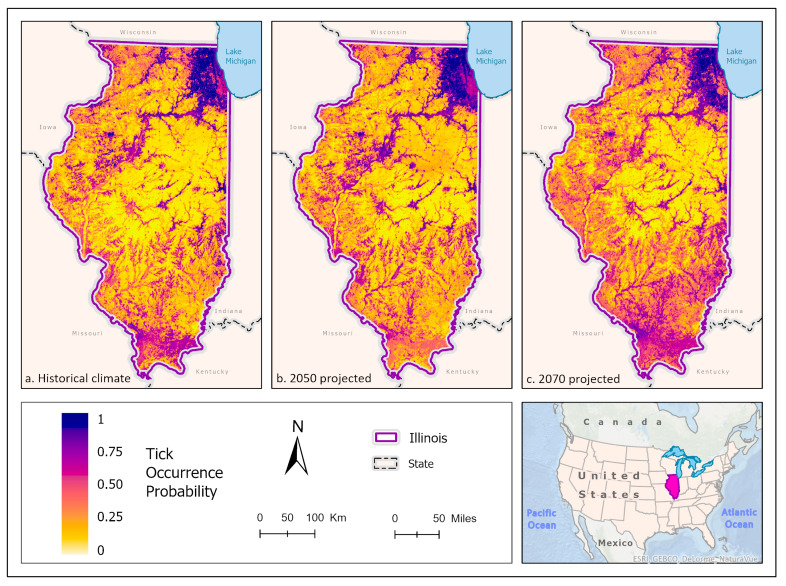
(**a**) Mean-weighted ensemble prediction of the probability of *I. scapularis* occurrence in Illinois under historical climate conditions. (**b**) Mean-weighted ensemble of predicted probability of *I. scapularis* occurrence in Illinois in 2050 projected climate Coupled Model Intercomparison Project phase 6 (CMIP6)/ EC-Earth3-Veg Shared Socioeconomic Pathway (SSP) 585 (average from 2041 to 2060). (**c**) Mean-weighted ensemble of future predicted probability of *I. scapularis* occurrence in Illinois in 2070 projected climate (CMIP6)/ EC-Earth3-Veg Shared Socioeconomic Pathway (SSP) 585 average from 2061 to 2080). Darker colors indicate a higher likelihood of tick presence, per the tick occurrence probability scale. Inset map indicates the location of Illinois within the United States/North America.

**Figure 3 insects-14-00213-f003:**
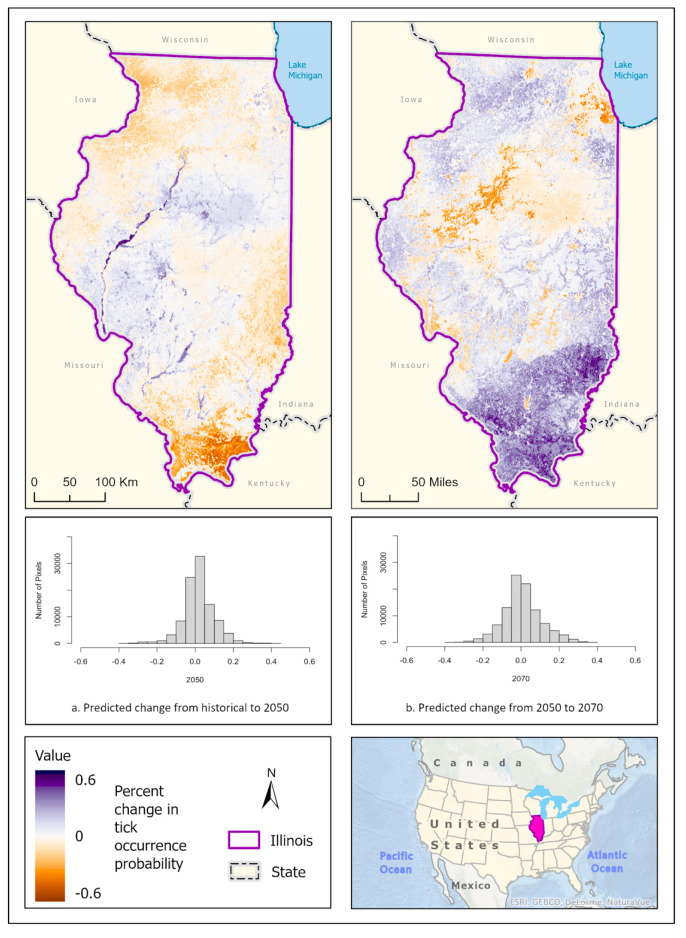
Percent change in the likelihood of *I. scapularis* occurrence between the historical climate and 2050 (**left**) and from 2050 to 2070 (**right**). Red shades indicate a reduced likelihood of occurrence (negative change), and blue shades indicate an increased likelihood of occurrence (positive change). The histograms represent the number of pixels (y−axis) containing the binned percentage likelihood change (x−axis) of *I. scapularis* suitable habitat across the map. Inset map indicates the location of Illinois within the United States/North America.

**Figure 4 insects-14-00213-f004:**
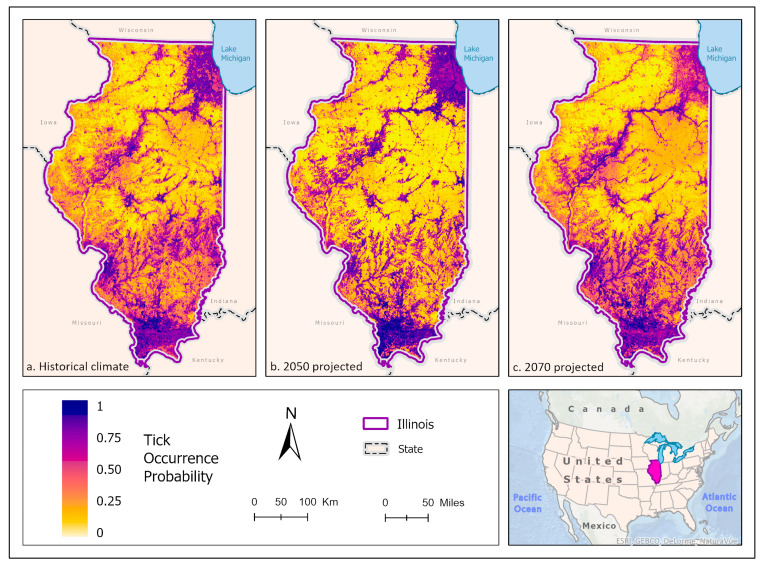
(**a**) Mean-weighted ensemble prediction of the probability of *A. americanum* occurrence in Illinois under historical climate conditions. (**b**) Mean-weighted ensemble of predicted probability of *A. americanum* occurrence in Illinois in 2050 projected climate Coupled Model Intercomparison Project phase 6 (CMIP6)/EC-Earth3-Veg Shared Socioeconomic Pathway (SSP) 585 (average from 2041 to 2060). (**c**) Mean-weighted ensemble of future predicted probability of *A. americanum* occurrence in Illinois in the 2070 projected climate (CMIP6)/EC-Earth3-Veg Shared Socioeconomic Pathway (SSP) 585 (average from 2061 to 2080). Darker colors indicate a higher likelihood of tick presence per the tick occurrence probability scale. Inset map indicates the location of Illinois within the United States/North America.

**Figure 5 insects-14-00213-f005:**
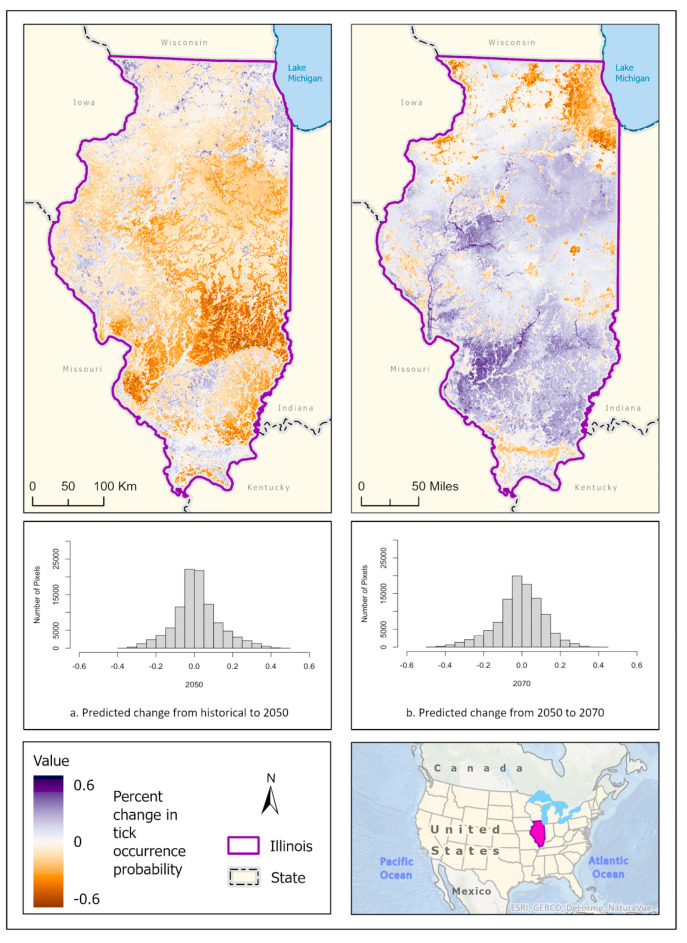
Percent change in the likelihood of *A. americanum* occurrence between the historical climate and 2050 (**left**) and from 2050 to 2070 (**right**). Red shades indicate a reduced likelihood of occurrence (negative change), and blue shades indicate an increased likelihood of occurrence (positive change). The histograms represent the number of pixels (y−axis) containing the binned percentage likelihood change (x−axis) of *A. americanum* suitable habitat across the map. Inset map indicates the location of Illinois within the United States/North America.

**Figure 6 insects-14-00213-f006:**
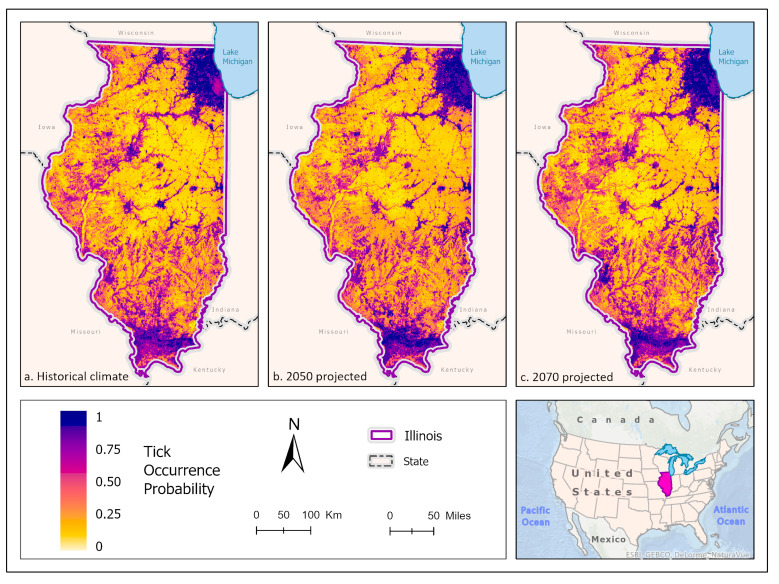
(**a**) Mean-weighted ensemble prediction of the probability of *D. variabilis* occurrence in Illinois under historical climate conditions. (**b**) Mean-weighted ensemble of predicted probability of *D. variabilis* occurrence in Illinois in 2050 projected climate Coupled Model Intercomparison Project phase 6 (CMIP6)/EC-Earth3-Veg Shared Socioeconomic Pathway (SSP) 585 (average from 2041 to 2060). (**c**) Mean-weighted ensemble of future predicted probability of *D. variabilis* occurrence in Illinois in 2070 projected climate (CMIP6)/EC-Earth3-Veg Shared Socioeconomic Pathway (SSP) 585 (average from 2061 to 2080). Darker colors indicate a higher likelihood of tick presence, per the tick occurrence probability scale. Inset map indicates the location of Illinois within the United States/North America.

**Figure 7 insects-14-00213-f007:**
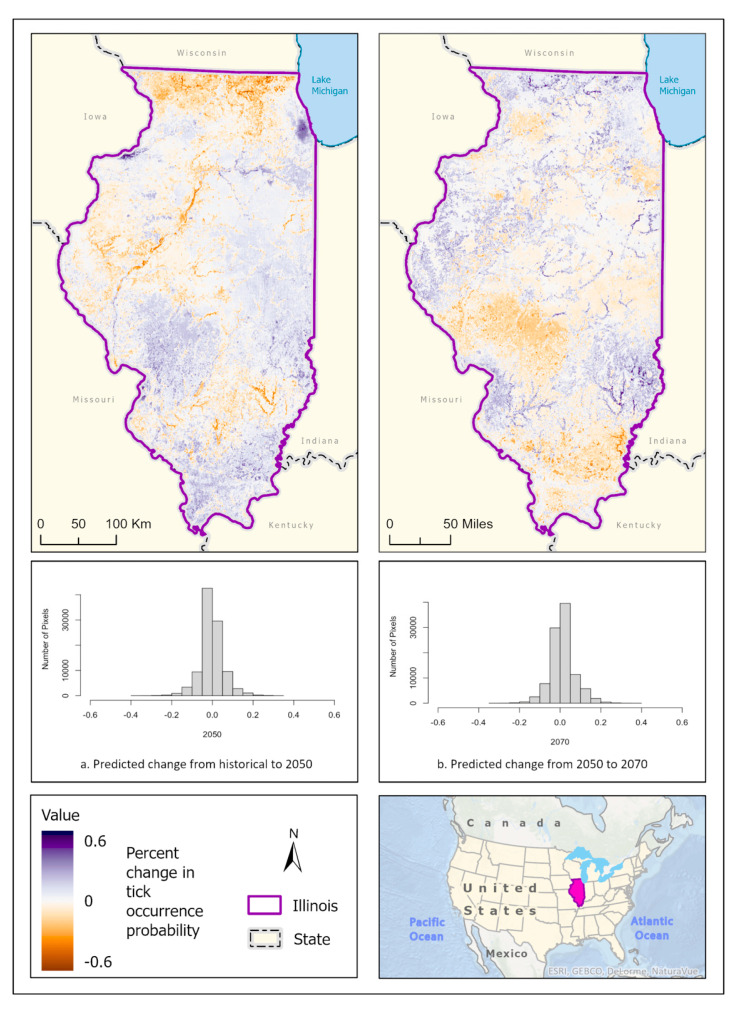
Percent change in the likelihood of *D. variabilis* occurrence between the historical climate and 2050 (**left**) and from 2050 to 2070 (**right**). Red shades indicate a reduced likelihood of occurrence (negative change), and blue shades indicate an increased likelihood of occurrence (positive change). The histograms represent the number of pixels (y−axis) containing the binned percentage likelihood (x−axis) of *D. variabilis* suitable habitat across the map. Inset map indicates the location of Illinois within the United States/North America.

**Figure 8 insects-14-00213-f008:**
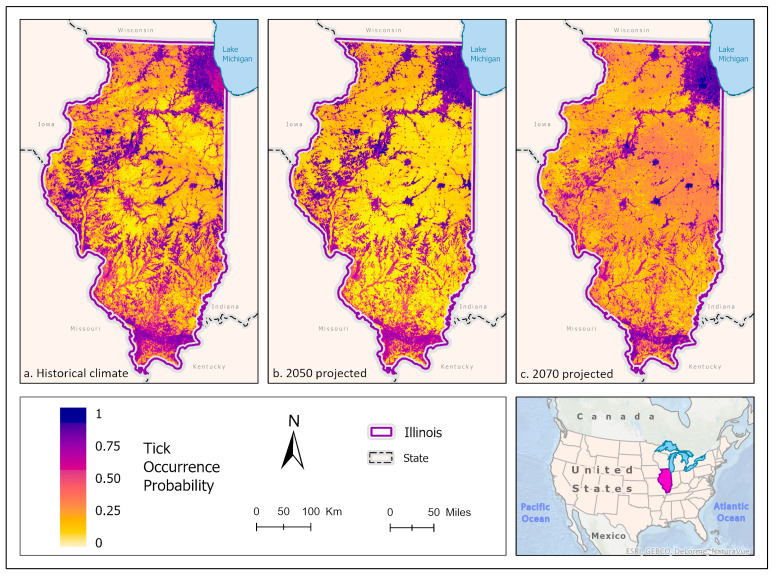
(**a**) Mean-weighted ensemble prediction of the probability of *A. maculatum* occurrence in Illinois under historical climate conditions. (**b**) Mean-weighted ensemble of predicted probability of *A. maculatum* occurrence in Illinois in 2050 projected climate Coupled Model Intercomparison Project phase 6 (CMIP6)/EC-Earth3-Veg Shared Socioeconomic Pathway (SSP) 585 (average from 2041 to 2060). (**c**) Mean-weighted ensemble of future predicted probability of *A. maculatum* occurrence in Illinois in 2070 projected climate (CMIP6)/EC-Earth3-Veg Shared Socioeconomic Pathway (SSP) 585 (average from 2061 to 2080). Darker colors indicate a higher likelihood of tick presence per the tick occurrence probability scale. Inset map indicates the location of Illinois within the United States/North America.

**Figure 9 insects-14-00213-f009:**
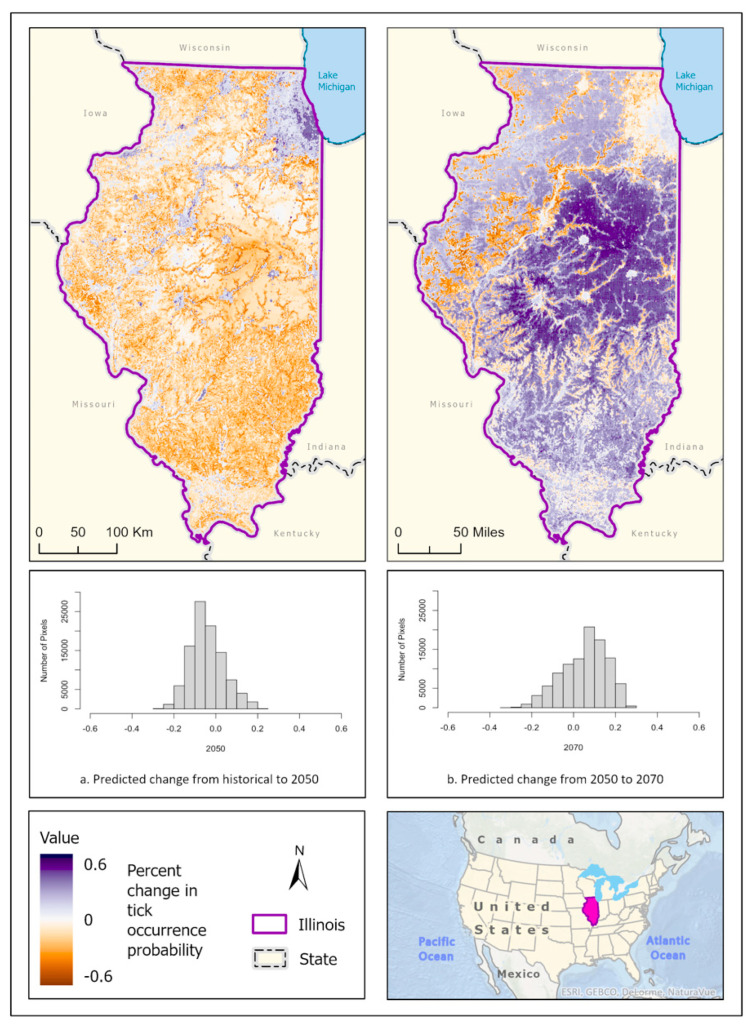
Percent change in the likelihood of *A. maculatum* occurrence between the historical climate and 2050 (**left**) and from 2050 to 2070 (**right**). Red shades indicate a reduced likelihood of occurrence (negative change), and blue shades indicate an increased likelihood of occurrence (positive change). The histograms represent the number of pixels (y−axis) containing the binned percentage likelihood (x−axis) of *A. maculatum* suitable habitat across the map. Inset map indicates the location of Illinois within the United States/North America.

**Table 1 insects-14-00213-t001:** Descriptions and sources of each of the 19 bioclimatic variables (WorldClim 2.1) and other environmental predictor variables (*n* = 29) used in model fitting.

VARIABLE	DESCRIPTION	Unit	Source *
BIO1	Annual Mean Temperature	°C	WorldClim 2.1
BIO2	Mean Diurnal Range (Mean of monthly (max temp—min temp))	°C	WorldClim 2.1
BIO3	Isothermality (BIO2/BIO7) (×100)	%	WorldClim 2.1
BIO4	Temperature Seasonality (standard deviation ×100)	°C	WorldClim 2.1
BIO5	Max Temperature of Warmest Month	°C	WorldClim 2.1
BIO6	Min Temperature of Coldest Month	°C	WorldClim 2.1
BIO7	Temperature Annual Range (BIO5-BIO6)	°C	WorldClim 2.1
BIO8	Mean Temperature of Wettest Quarter	°C	WorldClim 2.1
BIO9	Mean Temperature of Driest Quarter	°C	WorldClim 2.1
BIO10	Mean Temperature of Warmest Quarter	°C	WorldClim 2.1
BIO11	Mean Temperature of Coldest Quarter	°C	WorldClim 2.1
BIO12	Annual Precipitation	mm	WorldClim 2.1
BIO13	Precipitation of Wettest Month	mm	WorldClim 2.1
BIO14	Precipitation of Driest Month	mm	WorldClim 2.1
BIO15	Precipitation Seasonality	%	WorldClim 2.1
BIO16	Precipitation of Wettest Quarter	mm	WorldClim 2.1
BIO17	Precipitation of Driest Quarter	mm	WorldClim 2.1
BIO18	Precipitation of Warmest Quarter	mm	WorldClim 2.1
BIO19	Precipitation of Coldest Quarter	mm	WorldClim 2.1
Elevation	Height above sea level	m	USGS SRTM
Deer habitat	Suitable white-tailed deer habitat	Presence/absence	USGS GAP Analysis
Landcover CLASS	Water, developed, impervious, barren, forest, grassland, cropland, wetland	%	NLCD 2019

* WorldClim 2.1 [http://www.worldclim.com (accessed on 1 June 2022)], USGS SRTM [https://www.usgs.gov/centers/eros/science (accessed on 1 June 2022)], USGS Gap Analysis [https://gapanalysis.usgs.gov/apps/species-data-download/ (accessed on 23 November 2022)], NLCD [https://www.mrlc.gov/data/nlcd-2019-land-cover-conus (accessed on 1 June 2022)].

**Table 2 insects-14-00213-t002:** Mean best fit single model evaluation metrics for the predicted historic occurrence in Illinois of the four tick species modeled. Bolded numbers denote the AUC/correlation/true skill statistic (TSS) score/deviance for the best fit model for that species. To be included in ensemble models, an individual model must display an AUC of at least 0.75 and a TSS of at least 0.50. A dash indicates that an algorithm did not display a complete (100%) model run success percentage for that period.

	Species	*Ixodes scapularis*	*Amblyomma americanum*	*Dermacentor variabilis*	*Amblyomma maculatum*	*Ixodes scapularis*	*Amblyomma americanum*	*Dermacentor variabilis*	*Amblyomma maculatum*	*Ixodes scapularis*	*Amblyomma americanum*	*Dermacentor variabilis*	*Amblyomma maculatum*	*Ixodes scapularis*	*Amblyomma americanum*	*Dermacentor variabilis*	*Amblyomma maculatum*
	Number of occurrence (presence) points	62	99	290	15	62	99	290	15	62	99	290	15	62	99	290	15
	Number of pseudo-absence points	70	100	300	20	70	100	300	20	70	100	300	20	70	100	300	20
	**Evaluation** **Metric**	**AUC**				**Correlation**				**TSS**				**Deviance**			
**Algorithm ***	GLM	0.88	0.86	0.84	0.60	0.67	0.64	0.61	0.21	0.71	0.67	0.71	0.45	1.85	1.13	1.02	7.26
	BRT	0.89	0.89	0.85	-	0.68	0.68	0.63	-	0.72	0.72	0.63	-	0.99	1.00	1.04	-
	CART	0.81	-	0.81	0.70	0.57	-	0.56	0.36	0.59	-	0.57	0.47	1.46	-	1.21	2.68
	MaxEnt	0.88	0.87	0.87	0.75	0.66	0.64	0.65	0.43	0.71	0.68	0.65	0.64	0.96	0.97	0.93	1.38
	RF	**0.89**	**0.89**	**0.87**	0.76	**0.68**	**0.69**	**0.65**	0.43	**0.72**	**0.71**	**0.65**	0.61	**0.87**	**0.85**	**0.90**	1.29
	MARS	0.74	0.81	0.84	0.66	0.44	0.55	0.60	0.28	0.49	0.59	0.59	0.41	13.0	3.32	1.09	21.8
	SVM	0.89	0.86	0.86	**0.80**	0.68	0.64	0.64	**0.54**	0.72	0.66	0.63	**0.71**	0.88	0.94	0.94	**1.32**

* GLM = generalized linear models; BRT = Bayesian regression trees; CART = classification and regression tree; MaxEnt = maximum entropy; RF = random forest; MARS = multivariate adaptive regression splines; SVM = support vector machines.

**Table 3 insects-14-00213-t003:** Relative percent contribution of habitat suitability variables in best fit models for each tick species across the three climate scenarios. Current climate models were fit using the historical data representing the average climate measurements from 1970 to 2000 at a 1 km resolution. Future climate models were fit with mean projections of these data at a 1 km resolution using Coupled Model Intercomparison Project phase 6 (CMIP6)/ EC-Earth3-Veg Shared Socioeconomic Pathway (SSP) 8.5 for 2050 (average from 2041 to 2060) and 2070 (average from 2061 to 2080). All landscape variables represent the percentage of that landcover, except for elevation, which is measured in meters. The top three most important variables in the model prediction are bolded for each period. Variables that were not included in a model due to collinearity are denoted with a dash.

Environmental Variable *	Tick Species											
	*Ixodes scapularis*			*Amblyomma americanum*			*Dermacentor variabilis*			*Amblyomma maculatum*		
	Climate Scenario (SSP 8.5)											
	Hist.	2050	2070	Hist.	2050	2070	Hist.	2050	2070	Hist.	2050	2070
**Barren**	2.5	1.3	1.5	1.7	2.8	1.4	1.0	1.7	0.5	2.6	2.3	**9.8**
**Cropland**	1.0	1.2	1.5	1.9	1.3	4.5	0.7	0.6	0.8	10.0	3.1	8.6
**Developed**	0.7	1.5	0.9	-	-	-	1.6	1.2	0.8	**10.3**	**6.2**	7.6
**Deer Habitat**	-	-	-	**3.9**	2.1	2.4	-	-	-	-	-	-
**Elevation**	1.8	2.1	0.6	3.1	3.1	**6.2**	1.0	**3.8**	1.2	-	-	-
**Forest**	**16.5**	**27.2**	**28.0**	**11.8**	**21.8**	**11.8**	**30.4**	**26.5**	**30.0**	**29.1**	**50.0**	**45.3**
**Grass/Shrub**	**3.9**	**2.8**	1.7	0.5	1.0	0.7	0.8	0.5	0.4	2.3	1.6	7.7
**Water**	1.0	0.8	0.7	3.6	0.3	0.2	0.5	0.5	0.4	1.7	0.6	0.6
**Wetland**	**10.8**	**6.3**	**13.6**	**17.9**	**4.1**	**7.3**	**12.7**	**12.3**	**11.1**	**10.1**	**0.9**	**13.1**
**BIO2**	-	-	-	1.8	**4.2**	3.1	**2.5**	3.4	**4.2**	-	-	-
**BIO5**	1.5	1.7	2.1	-	-	-	-	-	-			-
**BIO7**	0.7	0.6	2.1	-	-	-	0.7	1.4	1.7	-	-	-
**BIO8**	1.5	0.5	0.9	0.9	1.1	0.4	1.1	1.1	0.6	-	-	-
**BIO9**	0.6	1.2	2.5	0.9	0.9	0.8	0.7	0.8	0.5	-	-	-
**BIO10**	-	-	-	-	-	-	0.8	0.7	0.6	-	-	-
**BIO13**	-	1.7	0.9	1.3	0.7	0.8	0.7	0.9	0.3	-	-	-
**BIO15**	-	-	-	2.4	2.9	1.2	-	-	-	2.8	3.9	**6.8**
**BIO18**	3.0	2.0	**3.0**	0.5	0.3	0.8	0.9	1.2	1.2	2.8	**7.8**	10.4

* BIO1, BIO3, BIO4, BIO6, BIO11, BIO12, BIO14, BIO16, BIO17, BIO19, and impervious surface were not included in any models due to multicollinearity issues among variables. BIO2 = mean diurnal range (mean of monthly (max temp—min temp)), BIO5 = max temperature of warmest month, BIO7 = temperature annual range (BIO5-BIO6), BIO8 = mean temperature of wettest quarter, BIO9 = mean temperature of driest quarter, BIO10 = mean temperature of warmest quarter, BIO13 = precipitation of wettest month, BIO15 = precipitation seasonality (coefficient of variation), BIO18 = precipitation of warmest quarter.

## Data Availability

Data and code can be accessed here: https://github.com/hkopsco/ILTickSDM.

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
