# Peer review of "Current and Future Habitat Suitability Models for Four Ticks of Medical Concern in Illinois, USA"

_insects, 2023, doi:10.3390/insects14030213_

Round 1
Reviewer 1 Report
This paper reports results of species distribution models for four species of ticks in Illinois, USA. The paper is well written; however, it is a bit too long, and importantly I have a few key comments regarding the methodology adopted especially with regard to land cover. Comments below:
A single model for future climate model (ACCESS 1.0) was used. However there exist numerous climate models on worldclim.org, and the outputs of each of these models would differ from each other to varying degrees. So, the recommended practice is to use an ensemble of multiple future climate models in species distribution modelling to reduce uncertainty. This is an important consideration and the omission needs to be justified.
Please justify the 1 km thinning distance when the resolution of the predictor sets is 4 km x 4 km. You would risk having multiple points in the same cell, thus there would be a perfect correlation among points leading to over or underestimation of the potential suitable habitat.
Line 175: The authors use the term ‘background’ points, but this is typically used in the context of MaxEnt where the idea is to sample the environment around the presence points. For other methods such as GLM, GAM, etc., in the species distribution modeling framework, the term “pseudo-absence” is used (since the true absences are often not available) as the authors are probably aware, and distance separation would ideally be the same as the presence points. Were these points thinned also as the presence points? In other words, were autocorrelations considered between “background” points and presence points?
For future climate scenarios the authors used the current land cover data (lines 211-216). The authors note on lines 157-159 that “Land cover changed significantly across the United States between 2001-2016..”, so it is surprising to me as to why the authors used the current land cover distribution for future time periods as the future land cover distribution (e.g., percent cropland, canopy cover, percent forest cover, etc.) will not be identical to the current situation. The authors touch upon this subject in the Discussion section (lines 723-730), but despite acknowledgement I think that this is a problem in the paper and renders the future projections inaccurate.
Line 184: A true skill statistic score of 0.4 is too low a bar for selecting models, I think.
Table 2. How would these models transfer to independent distribution records of the tick species studied? Since TSS scores on cross-validation test data are relatively low, I’m not sure how good these models are for classifying independent data.
Lines 502-509: Only 15 presence records for Amblyomma maculatum? The ratio of number of observations (35) to number of predictors (9) is too low.
Author Response
This paper reports results of species distribution models for four species of ticks in Illinois, USA. The paper is well written; however, it is a bit too long, and importantly I have a few key comments regarding the methodology adopted especially with regard to land cover. Comments below:
We have made many changes to this manuscript that partially resulted from finding an error in one of our raster layers. Once corrected, and after increasing the resolution of the analysis overall and addressing the helpful comments below, we believe that our paper is much improved. Many thanks to the reviewers for their thoughts and suggestions.
A single model for future climate model (ACCESS 1.0) was used. However there exist numerous climate models on worldclim.org, and the outputs of each of these models would differ from each other to varying degrees. So, the recommended practice is to use an ensemble of multiple future climate models in species distribution modelling to reduce uncertainty. This is an important consideration and the omission needs to be justified.
- Response: Thank you for this observation. Thank you for this comment. While multimodal means/ensembles for GCMs are the best practice in future climate projection, we were constrained by the R package utilized for this analysis so instead opted to use a model shown* to perform best in the region of our analysis (*see below for references). We’ve also updated our analysis to include the CMIP6 predictions (the previous version of this manuscript used CMIP5 GCMs). Additional limitation discussion has been added for this particular model.
Winkler, J.A., R.W. Arritt, S.C. Pryor. 2012: Climate Projections for the Midwest: Availability, Interpretation and Synthesis. In: U.S. National Climate Assessment Midwest Technical Input Report. J. Winkler, J. Andresen, J. Hatfield, D. Bidwell, and D. Brown, coordinators. Available from the Great Lakes Integrated Sciences and Assessment (GLISA) Center, http://glisa.msu.edu/docs/NCA/MTIT_Future.pdf.
- Ashfaq, D. Rastogi, J. Kitson, M.A. Abid, S.-C. Kao, Evaluation of CMIP6 GCMs over the CONUS for downscaling studies, J. Geophys. Res. 127 (2022). https://doi.org/10.1029/2022jd036659.
R.D. Bhowmik, A. Sharma, A. Sankarasubramanian, Reducing Model structural uncertainty in climate model projections—a rank-based model combination approach, J. Clim. 30 (2017) 10139–10154. https://journals.ametsoc.org/view/journals/clim/30/24/jcli-d-17-0225.1.xml?tab_body=abstract-display.
Please justify the 1 km thinning distance when the resolution of the predictor sets is 4 km x 4 km. You would risk having multiple points in the same cell, thus there would be a perfect correlation among points leading to over or underestimation of the potential suitable habitat.
Response: Thank you for catching this error. We have updated the resolution of the environmental variables to 30arcseconds (~1km) to fit the thinned point distances.
Line 175: The authors use the term ‘background’ points, but this is typically used in the context of MaxEnt where the idea is to sample the environment around the presence points. For other methods such as GLM, GAM, etc., in the species distribution modeling framework, the term “pseudo-absence” is used (since the true absences are often not available) as the authors are probably aware, and distance separation would ideally be the same as the presence points. Were these points thinned also as the presence points? In other words, were autocorrelations considered between “background” points and presence points?
Response: We have updated the terminology to better fit the conventions. Yes, autocorrelations were accounted for and all points were thinned using the same criteria.
For future climate scenarios the authors used the current land cover data (lines 211-216). The authors note on lines 157-159 that “Land cover changed significantly across the United States between 2001-2016..”, so it is surprising to me as to why the authors used the current land cover distribution for future time periods as the future land cover distribution (e.g., percent cropland, canopy cover, percent forest cover, etc.) will not be identical to the current situation. The authors touch upon this subject in the Discussion section (lines 723-730), but despite acknowledgement I think that this is a problem in the paper and renders the future projections inaccurate.
Response: The comment on landcover change was an acknowledgement of a note on the nationwide data by USGS, and served as the reason why we used a landcover raster that was an average across this period to account for all of the change. We’ve added additional acknowledgement that it would’ve been ideal to use future projected landcover in addition to future projected climate conditions, but since we could not match up model projections for both landcover and climate (i.e. the projections were made using different models), we did not want to include additional potential confounders and bias.
Line 184: A true skill statistic score of 0.4 is too low a bar for selecting models, I think.
Response: We’ve increased the score threshold to 0.5.
Table 2. How would these models transfer to independent distribution records of the tick species studied? Since TSS scores on cross-validation test data are relatively low, I’m not sure how good these models are for classifying independent data.
Response: With the changes to our analysis, the models have improved (evidenced by much higher TSS scores). We are confident that these models will handle independent data well.
Lines 502-509: Only 15 presence records for Amblyomma maculatum? The ratio of number of observations (35) to number of predictors (9) is too low.
Response: Unfortunately, these are all the presence records that exist that fit our criteria. We’ve added additional limitation information in the discussion.
Reviewer 2 Report
Greetings and Regards
The main research question
Current and future habitat suitability models for four ticks of medical concern in Illinois, USA
The introduction is properly described, I checked, but overall it needs more explanation.
The results of the paper, according to the authors, are predicted to contract in the 2050 climate scenario, but expand in the 2070 scenario. Anticipating where ticks may invade and concentrate due to climate change is important for predicting, preventing, and treating TBD in Illinois. And in general, in Sustainable Development of Geotechnical Engineering, it is suitable for publication in the Journal of Geotechnical Engineering
It is suggested to compare this species in other countries in the future.
Figure 1 is not clear and needs further explanation.
Table 2 needs a complete and understandable explanation. And it is also not of good quality.
In terms of written language and grammar, it needs changes.
In general, the article is suitable for publication with a few changes
Author Response
The introduction is properly described, I checked, but overall it needs more explanation.
Response: Thank you – additional details added to the introduction.
The results of the paper, according to the authors, are predicted to contract in the 2050 climate scenario, but expand in the 2070 scenario. Anticipating where ticks may invade and concentrate due to climate change is important for predicting, preventing, and treating TBD in Illinois. And in general, in Sustainable Development of Geotechnical Engineering, it is suitable for publication in the Journal of Geotechnical Engineering
It is suggested to compare this species in other countries in the future.
Response: We will made additional notes about the next steps for this research.
Figure 1 is not clear and needs further explanation.
Response: Further clarification made in the caption.
Table 2 needs a complete and understandable explanation. And it is also not of good quality.
Response: Thank you. Additional text added and the file resolution has been improved.
In terms of written language and grammar, it needs changes.
Response: We’ve made corrections where necessary.
In general, the article is suitable for publication with a few changes

Round 2
Reviewer 1 Report
I appreciate the effort of the authors in significantly revising the manuscript. Few minor corrections and clarifications are needed in this version.
Line 35: for all species?
Line 72: ‘I’ in lowercase.
Lines 75, 106, 107: Genus name repeated.
Lines 137-138: Is there an error here? CMIP6 and RCP, or is it SSP (“Shared Socio-economic Pathways”)? RCPs are scenarios from AR5 and precede SSPs.
Lines 164-165: include ° for the extent.
Line 196, Table 1. Were the units “°C*100” converted to "°C" prior to modelling and mapping? Unit for precipitation seasonality %?
Line 234, 237: "I. scapularis" in italics.
Line 237: ”they same”
Lines 271, 290, 352, 427, 511: “phase 5 (CMIP6)”? See line 137.
Line 307: “surface,”
Author Response
Response to Reviewers – Revision 2 (minor)
We are grateful for this thoughtful and careful review. We’ve made the noted changes to the manuscript, as well as updated some citations so that CMIP6/WorldClim 2 are properly documented.
Line 35: for all species?
Response: Updated
Line 72: ‘I’ in lowercase.
Response: Corrected
Lines 75, 106, 107: Genus name repeated.
Response: Corrected
Lines 137-138: Is there an error here? CMIP6 and RCP, or is it SSP (“Shared Socio-economic Pathways”)? RCPs are scenarios from AR5 and precede SSPs.
Response: Corrected to SSP 585. We simply missed this correction when we updated to CMIP6/ESM.
Lines 164-165: include ° for the extent.
Response: Updated.
Line 196, Table 1. Were the units “°C*100” converted to "°C" prior to modelling and mapping? Unit for precipitation seasonality %?
Response: Updated. We had conflicting unit sources and did not update the units in this table when we decided we didn’t need a scaling factor. The units for bio3 (Precipitation seasonality ratio) and bio15 (coefficient of variation) are listed in WorldClim as a percent, as well as here: https://pubs.usgs.gov/ds/691/ds691.pdf
Line 234, 237: "I. scapularis" in italics.
Response: Corrected.
Line 237: ”they same”
Response: Corrected.
Lines 271, 290, 352, 427, 511: “phase 5 (CMIP6)”? See line 137.
Response: All corrected.
Line 307: “surface,”
Response: Corrected.
